# Intermittent Fasting versus Continuous Calorie Restriction: Which Is Better for Weight Loss?

**DOI:** 10.3390/nu14091781

**Published:** 2022-04-24

**Authors:** Qing Zhang, Caishun Zhang, Haidan Wang, Zhengye Ma, Defeng Liu, Xiaohan Guan, Yixin Liu, Yanwen Fu, Mingxuan Cui, Jing Dong

**Affiliations:** 1Special Medicine Department, Medical College, Qingdao University, Qingdao 266071, China; 2020021152@qdu.edu.cn (Q.Z.); 2019010076@qdu.edu.cn (C.Z.); 2019021021@qdu.edu.cn (H.W.); 2Clinical Medicine Department, Medical College, Qingdao University, Qingdao 266071, China; 2017210118@qdu.edu.cn (Z.M.); 2017210161@qdu.edu.cn (D.L.); 2019210499@qdu.edu.cn (X.G.); 2019210569@qdu.edu.cn (Y.L.); 2019210442@qdu.edu.cn (Y.F.); 2019410003@qdu.edu.cn (M.C.); 3Physiology Department, Medical College, Qingdao University, Qingdao 266071, China

**Keywords:** intermittent fasting, calorie restriction, energy control, BMI, body weight, meta-analysis

## Abstract

We conducted a systematic review and meta-analysis of randomized clinical trials and pilot trial studies to compare the effectiveness of intermittent fasting (IF) and continuous calorie restriction (CCR) in overweight and obese people. The parameters included body mass index (BMI), body weight, and other metabolism-related indicators. A systematic search in PubMed, Embase, Cochrane Library, and Web of Science was conducted up to January 2022. Standardized mean differences (SMDs) with 95% confidence intervals (CIs) were used to measure the effectiveness. Publication bias was assessed using Egger’s test. The stability of the results was evaluated using sensitivity analyses. The significance of body weight change (SMD = −0.21, 95% CI (−0.40, −0.02) *p* = 0.028) was more significant after IF than CCR. There was no significant difference in BMI (SMD = 0.02, 95% CI (−0.16, 0.20) *p* = 0.848) between IF and CCR. These findings suggest that IF may be superior to CCR for weight loss in some respects.

## 1. Introduction

Compared to 173 million obese people in 2014, 257 million adults worldwide (6% of men and 9% of women) are predicted to be living with severe obesity, showing a rapid increase in the number of obese people [1]. Obesity is now replacing malnutrition and infectious diseases as the most critical cause of suboptimal health. Obesity has been linked to diabetes, coronary heart disease, cancer, metabolic syndrome, and sleep-disordered breathing [2]. In addition, obesity can also cause the elevation of oxidative stress, inflammatory state, and hypoxia, which leads to the dysfunction of perivascular adipose tissue (PVAT) [3,4].

Metabolic syndrome is also a growing concern characterized by pathological metabolism of protein, fat, carbohydrates, and other substances; it is a risk factor for diabetes, cardiovascular, and cerebrovascular diseases [5]. The amount and quality of many types of cells found in adipose tissue, including adipose stem cells (ASC), is altered as a result of obesity. These changes in the function and nature of ASC impair adipose tissue remodeling and adipose tissue function, leading to metabolic disorders [6]. The common causes of these diseases are insulin resistance and hyperinsulinemia secondary to obesity, especially central obesity [7].

Most obese people rarely choose to exercise because of work stress or psychological reasons; they are more enthusiastic about dietary intervention. Dietary adjustment is the heart of obesity treatment. Weight loss diets include various permutations of energy restriction, macronutrients, and food and dietary intake patterns [8]. In recent years, various dietary adjustment methods have become increasingly popular. Current guidelines recommend continuous calorie restriction (CCR; about 500 or 750 kcal of energy deficiency per day, or 30% of baseline energy requirements limits) and comprehensive lifestyle interventions as the cornerstone of obesity treatments [9]. On average, this method produces moderate weight loss (5–10% ≥ 1 year) [9].

Because traditional CCR methods are relatively ineffective in achieving and sustaining weight loss, there has been growing interest in identifying alternative dietary weight-loss strategies that limit the energy intake to specific periods of the day or extend the gap between meals (i.e., intermittent calorie restriction, ICR) [10]. Intermittent fasting (IF) comes in many forms and includes regular breaks. A common form of IF includes fasting once or twice a week for up to 24 h, followed by discretionary food intake for the remainder of the day, known variously as periodic long-term fasting, ICR, intermittent energy restriction (IER) [11], time-restricted eating (TRE, i.e., eating only for 8 h and then fasting for another 16 h a day), and alternate-day fasting (ADF) [12]. However, it is not yet certain whether TRE has the same health effects as other forms of IF [10,13]; therefore, we omitted TRE in this study.

Continuous fasting has become widespread in daily life, and intermittent fasting has become increasingly popular. Intermittent fasting is a dietary pattern alternating between normal energy and energy restriction (or complete fasting) and has attracted substantial attention from scholars [14]. IF reduces body mass and improves glucose and lipid metabolism. Its benefits include reducing the risk of diabetes, cardiovascular disease, and stroke, inhibiting tumor growth, and preventing Alzheimer’s disease and Parkinson’s disease [15].

A study discovered that IF might be comparable to, but not superior to, CCR for weight loss and metabolic illness prevention [10]. However, another study suggested that IF is a more effective strategy in managing the body weight, fat mass, and waist circumference of individuals with metabolic syndrome [16]. These authors found that, compared with CCR, IF decreased the levels of high-sensitivity C-reactive protein and prothrombin and thromboplastin times. A review suggested that IF might be superior to CCR because it helps conserve lean body mass at the expense of fat mass [17]. A growing body of evidence suggests that IF has more benefits and can be more effective than CCR. Nevertheless, IF versus CCR on weight loss in overweight and obese people remains controversial.

At present, there are a variety of obesity evaluation indicators in the home and abroad; the most commonly used are: body mass index (BMI in kg/m^2^), body fat rate (body fat percentage, BF%), waist circumference (WC), body weight (BW), etc. Among them, BMI is widely used to evaluate generalized obesity. The BF% is widely applied to evaluate the proportion of body fat, WC is widely used to evaluate valence abdominal obesity, and BW is mainly used to evaluate abdominal obesity and health risk. Therefore, our research mainly looked at the BMI, BW, WC, etc.

Because of the growing concern regarding obesity and the diseases it causes, a literature review was carried out to compare these weight-loss strategies. Therefore, our study aimed to compare IF and CCR regarding effectiveness for weight loss in people with obesity or metabolic syndrome.

## 2. Materials and Methods

Our meta-analysis was based on the Preferred Reporting Items for Systematic Reviews and Meta-analyses guidelines [18].

### 2.1. Data Sources and Search Strategy

Four significant websites were used to compile our data (PubMed, Embase, Cochrane Library, and Web of Science). The search terms used for the studies were two sets of keywords and their main subtitles, including calorie restriction (“intermittent calorie restriction”, “intermittent energy restriction”, “intermittent fasting”, and “alternate day fasting”) and BMI (“body mass index”, “body weight”, “weight loss”, “weight gain”, “fat mass”, “obesity”, “overweight”, “insulin-resistance”, and “insulin sensitivity”). The date range was 1 January 2000, to 1 February 2022. More details are shown in Figure 1.

### 2.2. Study Selection and Criteria

Studies were eligible if they met the following criteria: (1) studies of adults (≥18 years old) with overweight or obesity based on BMI (≥25 kg/m^2^); (2) studies comparing nutritional interventions for weight loss based on IF; (3) studies comparing nutritional interventions for weight loss based on CCR; (4) studies describing body weight loss and modifications of body composition; (5) studies published in English; (6) subjects of original articles ≥10.

Exclusion criteria were as follows: (1) studies with unreliable designs or substantial statistical errors; (2) only one type of diet regimen included; (3) inability to access the full text. 

### 2.3. Data Extraction and Quality Assessment

Two independent reviewers reviewed all the studies and extracted the data in a standardized format. The articles in the database were retrieved based on the data measured at the end of each study, and we extracted the most complete and recent data based on the articles from the same population. The information collected was as follows: first author, publication year, country, study design, participants, study duration, age, BMI, body weight, fat mass, waist, fasting blood glucose, and interventions. The data were converted to unified units as needed. We did not include interventions such as exercise and drugs to avoid affecting the effectiveness of our evaluation.

We compared IF with CCR, without restriction on the treatment history. The outcomes were as follows: change in BMI between baseline and the end of intervention, and change in body weight between baseline and the end of the intervention.

For randomized clinical trials, we used the Cochrane methodology to assess the quality [19] (Appendix A).

### 2.4. Data Synthesis and Analysis

The comparison between IF and CCR was analyzed using the random-effects model, which used mean values and standard deviations. Continuous variables were analyzed using standard mean difference (SMD) and 95% confidence intervals (CIs). We utilized Cochran’s (chi-square) test to measure heterogeneity and the I^2^ statistic to determine the extent of consistency: an I^2^ of over 75% indicates a high level of inconsistency, I^2^ of above 50% is moderate, and I^2^ of below 25% is low [20]. To estimate pooled effect sizes, random effects models were used. A two-sided *p*-value of less than 0.05 was considered statistically significant. Publication bias was assessed using Begg’s test [21] and Egger’s test [22].

We conducted influence analysis to determine the impact of a single study on the overall results. Subgroup analysis was performed according to the intermittent fasting regimen, age, and area. All statistical analyses were carried out using RevMan (Version 5.3) and Stata Software (Version 12.0).

## 3. Results

### 3.1. Literature Search

The details of the search are shown in Figure 1. A total of 10,138 potential reports were identified from PubMed, Embase, Cochrane Library, and Web of Science. Based on the year of publication and the type of study, we included 1971 articles. After re-examination, 304 duplicated studies were deleted. After reviewing the titles and abstracts, 1615 nonconforming studies were excluded. After reviewing the full texts, 41 studies that did not meet the requirements were excluded. Finally, eleven studies met the selection criteria [23,24,25,26,27,28,29,30,31,32,33].

### 3.2. Study Characteristics

We included 11 articles with 705 patients. The studies’ characteristics are displayed in Table 1 and Table 2. The 11 studies were primarily randomized controlled trials. The BMIs of all participants were more than 25 kg/m^2^, and some of the patients were diagnosed with type 2 diabetes mellitus or metabolic syndrome. Eight studies compared IF with CCR [23,25,26,27,28,29,31,33], two compared ADF with CCR [30,32], one compared CCR with twice-weekly fasting (i.e., on any two non-consecutive days of the week, girls only ate 500 kcal per day and boys only ate 600 kcal per day. On the other five days, the subjects ate normally, but did not overeat or diet) [24]. ICR, ADF, and twice-weekly fasting are all forms of IF. All studies included data on BMI and weight. All dietary intervention methods met the standard criteria, and data measurements were ensured to minimize errors. The studies included regular follow-up to ensure accuracy. All subjects were above 30 years old.

### 3.3. Meta-Analysis

#### 3.3.1. BMI after IF versus CCR

There was no significant difference between IF and CCR for BMI (SMD = 0.02, 95% CI (−0.16, 0.20) *p* = 0.848; Figure 2). The SMD showed no significant heterogeneity using the random-effect model (I^2^ = 45.0%, *p* = 0.046). Publication bias was insignificant (Appendix A; Egger’s test: *p* = 0.730). The funnel plot is shown in Figure 3.

#### 3.3.2. Body Weight after IF Versus CCR

There was a significant difference between IF and CCR for weight (SMD = −0.21, 95% CI (−0.40, −0.02) *p* = 0.028; Figure 4). The SMD showed no significant heterogeneity using a random-effect model (I^2^ = 48.1%, *p* = 0.031). Publication bias was insignificant (Appendix A; Egger’s test: *p* = 0.401). The funnel plot is shown in Figure 5.

#### 3.3.3. TC, TG and Waist Circumference after IF versus CCR

We also analyzed the total cholesterol, triacylglycerol, and waist circumference. There was a significant difference between IF and CCR for the total cholesterol (SMD = −0.06, 95% CI (−0.26, 0.14) *p* = 0.538; Figure 6). The SMD showed no heterogeneity using a random-effect model (I^2^ = 0%, *p* = 0.502). There was a significant difference between IF and CCR for triacylglycerol (SMD = −0.12, 95% CI (−0.32, 0.08) *p* = 0.252; Figure 7). The SMD showed no heterogeneity using a random-effect model (I^2^ = 0%, *p* = 0.855). There was a significant difference between IF and CCR for waist circumference (SMD = −0.10, 95% CI (−0.41, 0.20) *p* = 0.508; Figure 8). The SMD showed no heterogeneity using a random-effect model (I^2^ = 58.8%, *p* = 0.024).

### 3.4. Subgroup Analysis

#### 3.4.1. Subgroup Analysis of BMI

We conducted a subgroup analysis because of the moderate heterogeneity of BMI change in the overall analysis. We found that the different IF forms accounted for the heterogeneity between IF and CCR. Therefore, all studies were divided into modified ADF and normal IF. Two studies with modified ADF showed significant differences between IF and CCR (SMD = −0.56, 95% CI: −0.90 to −0.23, I^2^ = 0%, *p* = 0.709), while the remaining studies with normal IF showed no significance (SMD = 0.14, 95% CI: −0.01 to 0.28, I^2^ = 0.0%, *p* = 0.059) (Figure 9). Because both sets of heterogeneity were zero, we deduced that the form of IF was the source of heterogeneity.

We carried out other subgroup analyses classified by age (≥60 y, <60 y), area (Oceania, Europe, Western Asia, North America), and physical condition (obesity or overweight, obesity or overweight with disease) (Table 3). These factors showed no significant differences or reductions in heterogeneity. More information is needed for further analysis.

#### 3.4.2. Subgroup Analysis of Body Weight

Because of the moderate heterogeneity of body weight change in the overall analysis, we conducted a subgroup analysis. We performed the same analysis as the BMI sub-group research. Two studies of modified ADF fasting showed that IF was more effective than CCR (SMD = −0.69, 95% CI: −1.02 to −0.23, I^2^ = 0%, *p* = 0.357). Nine studies of normal IF showed no significant difference between IF and CCR (SMD = −0.10, 95% CI: −0.26 to 0.05, I^2^ = 13.5%, *p* = 0.319) (Figure 10).

We carried out other subgroup analyses classified by age (≥60 y, <60 y), area (Oceania, Europe, Western Asia, North America), and physical condition (obesity or overweight, obesity or overweight with disease) (Table 4). No statistical significance was found for these factors.

#### 3.4.3. Publication Bias and Sensitivity Analysis

Egger’s test was used to measure publication bias (*p* = 0.730 for BMI and *p* = 0.401 for body weight). Meta-analysis of articles included in this study revealed no publication bias (Figure 11 and Figure 12). The stability of our meta-analysis was demonstrated by the sensitivity analysis (Figure 13 and Figure 14).

## 4. Discussion

We performed a systematic review and meta-analysis comparing IF and CCR regimens regarding BMI and body weight reduction. IF is beneficial for weight loss, and the effect was significant. If dietary intervention is well studied, it will substantially impact society. Our findings are consistent with those of Enríquez Guerrero et al. [34].

Calorie restriction involves reduced caloric intake of about 25–30% without eliminating essential nutrients [35]; this approach prolongs health and life in rodent and primate models [36,37]. The mechanisms of these benefits are related to the inhibition of anabolism, improvement of mitochondrial energy metabolism, and the conversion of substrate utilization. These processes are related to the reduced dependence on glucose metabolism and increased fatty acid oxidation [35]. IF is an effective dietary intervention because it improves the lipid profile and reduces body weight [38]. IF has a positive effect on glycolipid metabolism in obese individuals. A study showed that eight consecutive weeks of ADF in obese adults led to a 6.8% reduction in blood glucose levels after fasting and a 22.6% reduction in insulin concentrations [12]. ADF improved insulin signaling and altered the proportion of α and β cells in obese mouse pancreases by reducing β cell apoptosis, increasing Akt (serine/threonine) phosphorylation, and improving diet-induced obesity islet tissue remodeling and β cell function [39]. IF improved glucose homeostasis through autophagy in a rodent model; TRE promoted the expression of glycolytic genes (Hk2, PFK, and PK) in obese mice and inhibited the expression of gluconeogenesis (G6pc, Pck1, and Fbp1), thereby promoting glucose uptake in peripheral tissues, and inhibited gluconeogenesis, ultimately reducing the blood glucose levels [40]. IF regulates glucose homeostasis by the intestinal flora [41]. ADF improves blood lipid levels whilst reducing the body mass and body weight, related to the depletion of liver glycogen reserves during fasting; triglyceride levels indicated that free fatty acids are released into liver cells to produce ketone energy [42]. IF (1 day of fasting followed by 2 days of feeding) promotes browning of white adipose tissue; the possible mechanisms include the activation of type II cell signaling through increasing the secretion of IL-5, stimulation of M2 macrophages, and reduction of M1/M2 macrophage ratios [43].

In addition to its positive effects of fasting, side effects are inevitable, such as muscle pain, sleep disturbances, headaches, and hunger, occurring mainly in the first few days of fasting [44]. In the group with longer fasting periods, baseline values for emotional well-being (EWB) and physical well-being (PWB) were lower [44]. These side effects of fasting do not occur for all people because of different physical fitness levels, including personal health, physiological mechanisms, lifestyle, etc.

Identifying long-term effective dietary interventions is critical to reducing the range of diseases caused by obesity [45]. Many people lead sedentary lifestyles because of office work, and exercise may be challenging. Therefore, dietary interventions are becoming more popular.

Elevated BMI is correlated with disease prevalence, suggesting that reducing BMI will reduce the disease burden. Epidemiological studies showed that elevated BMI could cause cardiovascular disease, diabetes mellitus, chronic kidney disease [46,47], cancers [48], and musculoskeletal disorders [49,50]. In the studies we considered, all subjects had BMIs greater than 25 kg/m^2^. Our meta-analysis found that IF and CCR improved BMI, and weights decreased significantly. While our analysis showed no difference between the two interventions for improvement in BMI, we can draw some inferences from the analysis. Of the eleven studies, seven showed a relatively significant decrease in BMI with CCR [23,24,25,28,29,31,33], and four showed a relatively significant decrease with IF [26,27,30,32]; however, the differences in these comparisons were relatively small. In the analysis of BMI, heterogeneity was 45% (moderate). The source of heterogeneity was different fasting days of IF. Therefore, we suspect that the weight loss effect is more significant when the number of fasting days in a week is greater than two.

We considered age as a possible source of heterogeneity. The heterogeneity of three studies, including subjects older than 60, was zero. Age is correlated with BMI to some extent [51]; as people age, their BMI increases. We speculate that CCR may be more appropriate in older people, especially the elderly. Their autoimmunity is relatively low, they require food to replenish energy, and their ability to endure hunger is relatively weak. Because their bodies have poor metabolic capacity, extremely low-calorie restriction throughout the day can be harmful. Therefore, persistent calorie restriction is a safer dietary intervention for them. This view was also mentioned in a review [52]. Studies showed that ADF did not produce superior adherence to daily calorie restriction [53]. In previous studies on other forms of IF, abandonment was reported in up to 40% of participants [54]. Therefore, future studies should determine the appropriate dietary intervention method according to the population and individual wishes.

We also analyzed changes in body weight. As articulated in the results, the differences in weight between the two interventions showed significance. The analysis showed that IF was more effective for weight loss. This finding suggests that IF requires further study and in-depth exploration. The weight loss effect of IF has been demonstrated in clinical trials. A strict IF of 4 to 24 weeks reduced subject body mass by 4–10% [55]. Other scholars pointed out that the role of IF in weight loss is not significantly different from that of a calorie-restricted diet, consistent with our analysis; however, IF was better at maintaining lean body mass [17].

11 May was declared World Obesity Day by the World Health Organization. Today, obesity is no longer merely a threat to individual health; it is also a social problem that has attracted worldwide attention for over 30 years, as overweight and obesity continues to grow. All sectors of society should work together to create a healthy environment that supports the active adoption of healthy lifestyles.

IF and CCR have powerful weight loss effects. Although weight and fat mass decreased in most studies, it is crucial to consider protocol adherence and exit rates. Sundfør et al. showed that IF subjects were hungrier than CCR subjects, and their willingness to persist decreased [56]. However, IF also offers some benefits. There are several studies on the effects of IF on cardiovascular disease in humans. A rat study found that IF improved glycemic control and protected the myocardium from ischemia-induced cell damage and inflammation more than daily CR [57]. These findings suggest that IF has substantial clinical significance, and it is a dietary intervention method that deserves in-depth research.

Our meta-analysis has some limitations. Firstly, the sample sizes of some of the included studies were small, leading to heterogeneity. More largescale studies are necessary to enhance the accuracy of our meta-analysis. Second, the follow-up time varied widely, and some did not follow-up.

In our study, it seems that ADF produces better results. More research is therefore needed to assess the mechanism of intermittent fasting regimens, and the safety of each type of them is also to be valued to finally determine applicable dietary interventions for specific groups of people.

In summary, we found that IF was more effective than CCR for weight loss; however, there was no difference in BMI improvement. Although the data are insufficient, our study shows that IF is superior to CCR in metabolism in obese people. We hope that there will be more long-term studies of dietary interventions and further investigation on cognitive function, which may reduce the economic burdens caused by obesity. Studies need to compare IF and CCR with controlled patient characters to confirm the effectiveness of these weight-loss methods and to determine whether IF is more appropriate for specific populations.

## Figures and Tables

**Figure 1 nutrients-14-01781-f001:**
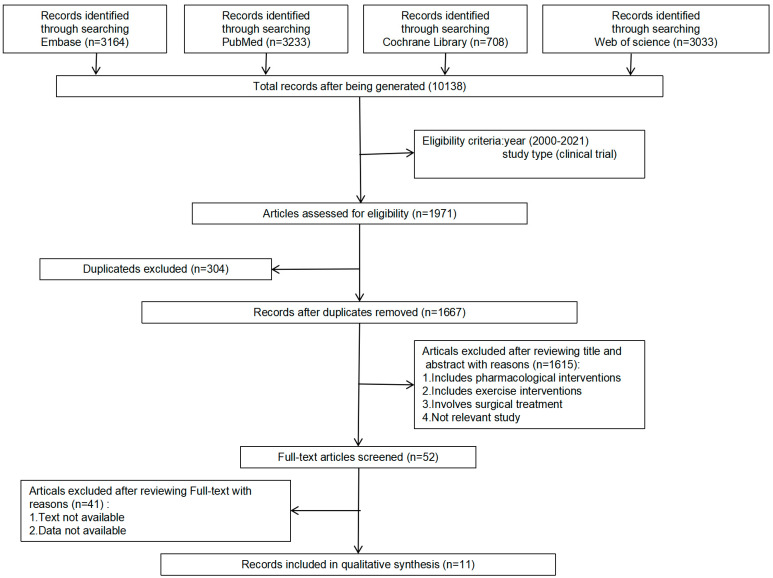
Flow chart of literature search.

**Figure 2 nutrients-14-01781-f002:**
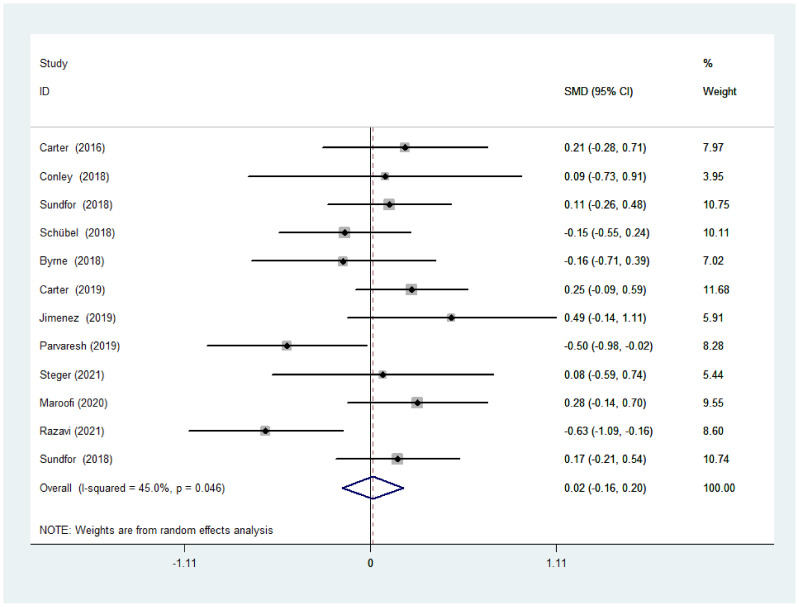
The weighted mean difference in BMI (kg/m^2^) between the IF and CCR (SD: standard deviation; CI: confidence interval).

**Figure 3 nutrients-14-01781-f003:**
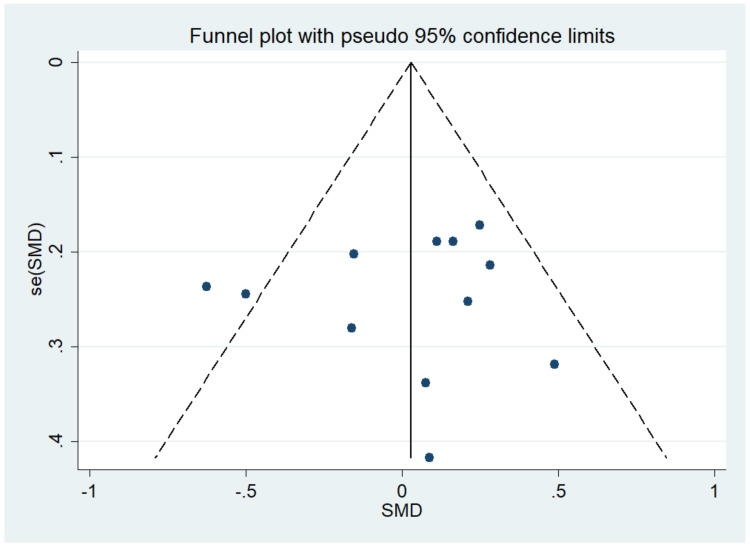
The funnel plot of the publication bias of BMI.

**Figure 4 nutrients-14-01781-f004:**
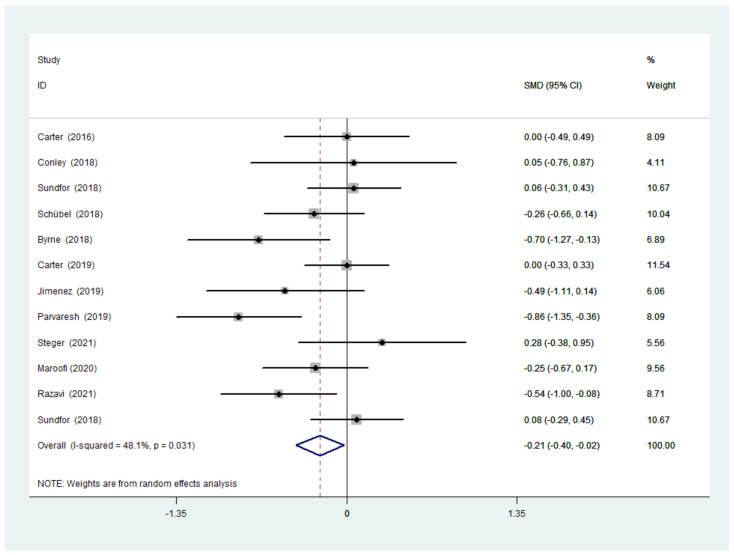
The weighted mean difference in body weight (kg) between the IF and CCR (SD: standard deviation; CI: confidence interval).

**Figure 5 nutrients-14-01781-f005:**
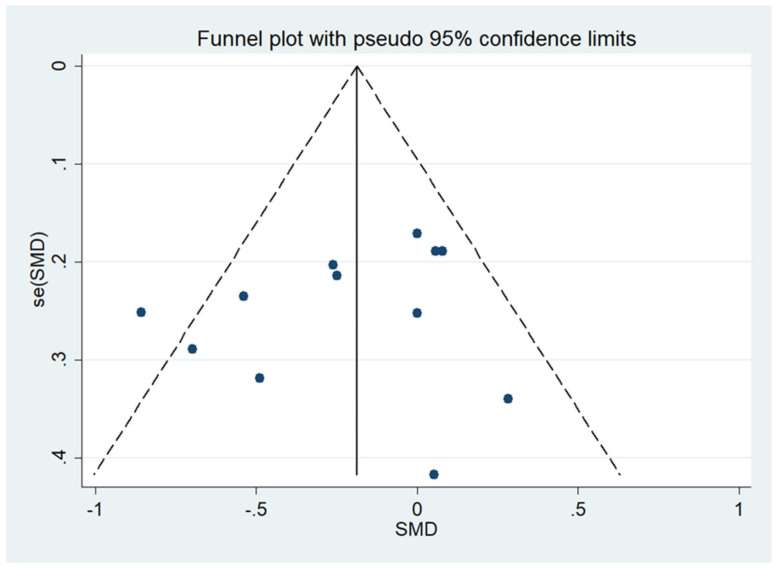
The funnel plot of the publication bias of body weight.

**Figure 6 nutrients-14-01781-f006:**
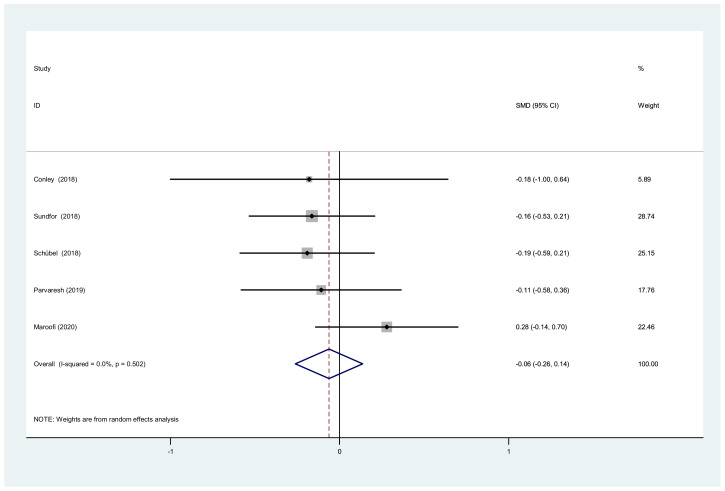
Forest plot of meta-analysis on comparing TC changes after IF and CCR interventions.

**Figure 7 nutrients-14-01781-f007:**
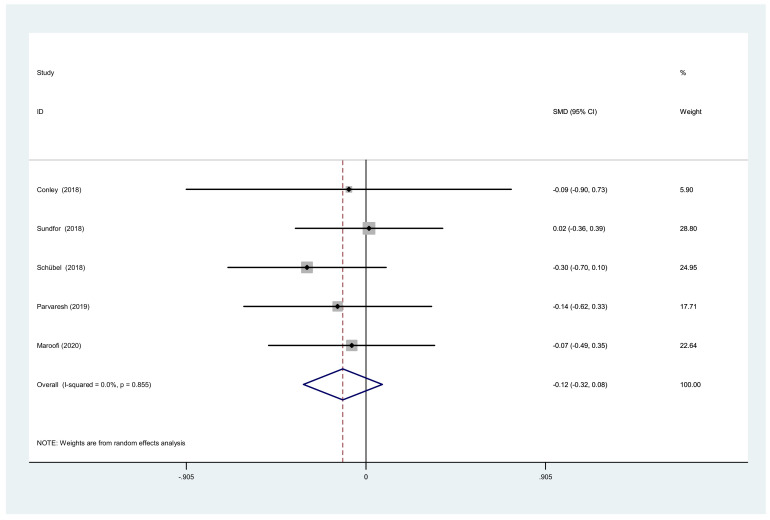
Forest plot of meta-analysis on comparing TG changes after IF and CCR interventions.

**Figure 8 nutrients-14-01781-f008:**
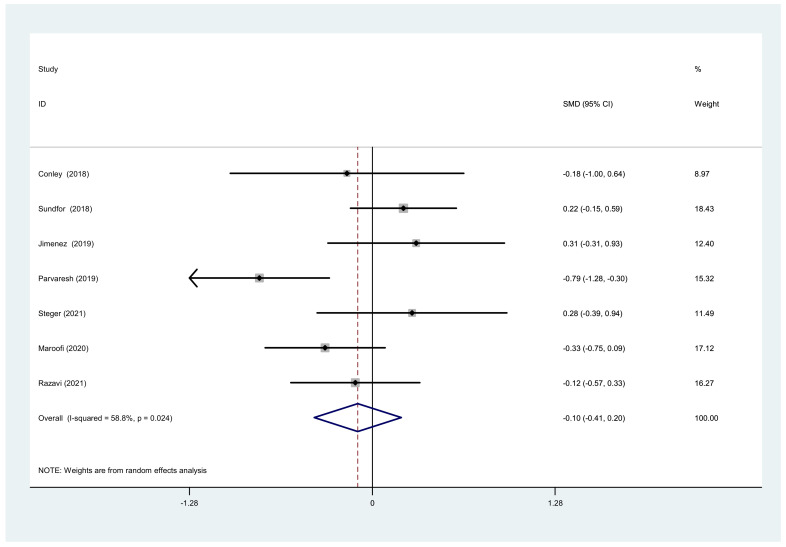
Forest plot of meta-analysis on comparing waist changes after IF and CCR interventions.

**Figure 9 nutrients-14-01781-f009:**
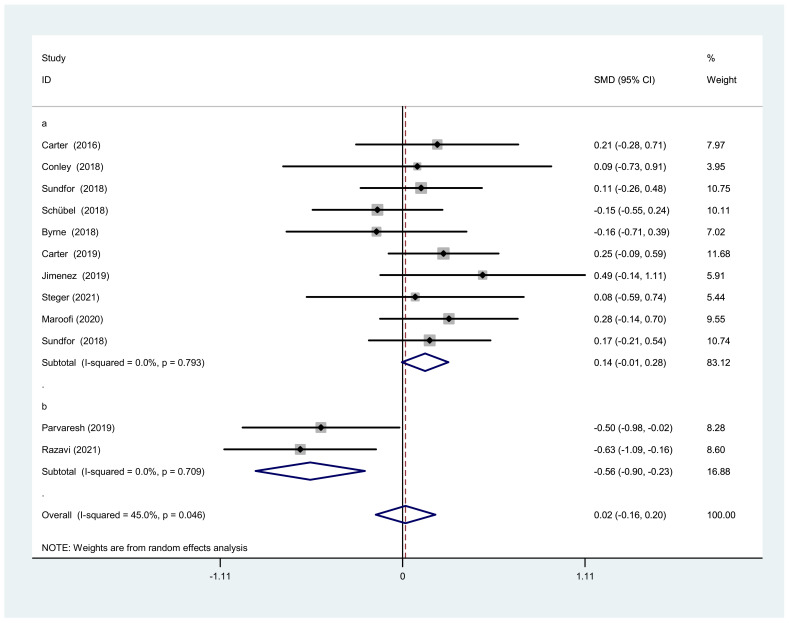
Forest plot of BMI changes after IF and CCR interventions based on fasting days. (**a**) Normal intermittent fasting (calorie restriction is usually two days per week). (**b**) Modified alternate-day fasting (a very lowcalorie diet (75% energy restriction) during the three fast days).

**Figure 10 nutrients-14-01781-f010:**
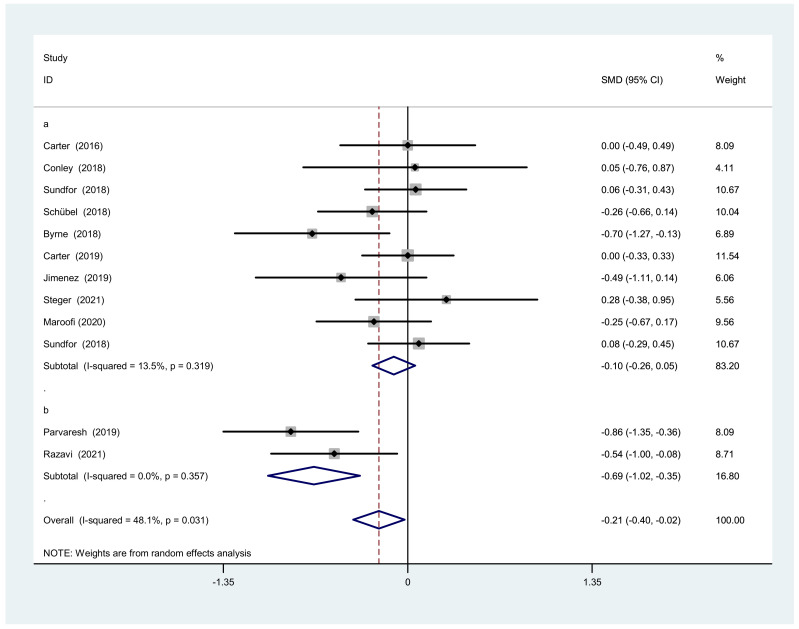
Forest plot of body weight changes after IF and CCR interventions based on fasting days. (**a**) Normal intermittent fasting (calorie restriction is usually two days per week). (**b**) Modified alternate-day fasting (a very low-calorie diet (75% energy restriction) during the three fast days).

**Figure 11 nutrients-14-01781-f011:**
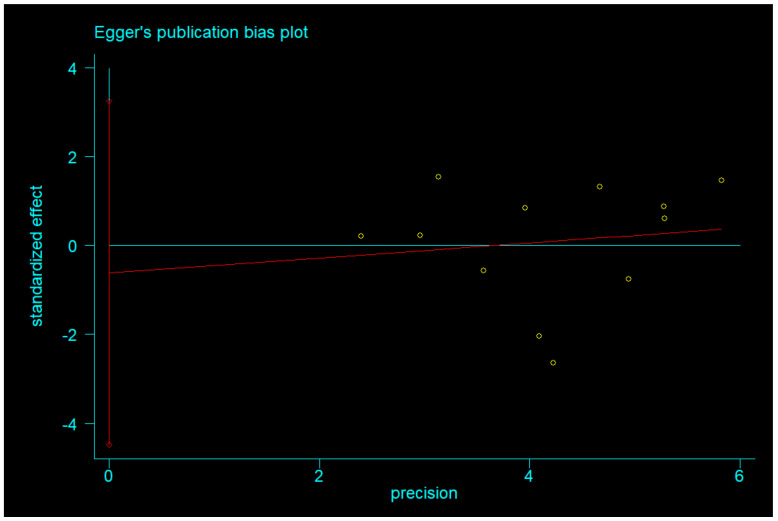
Egger’s publication bias plot of BMI.

**Figure 12 nutrients-14-01781-f012:**
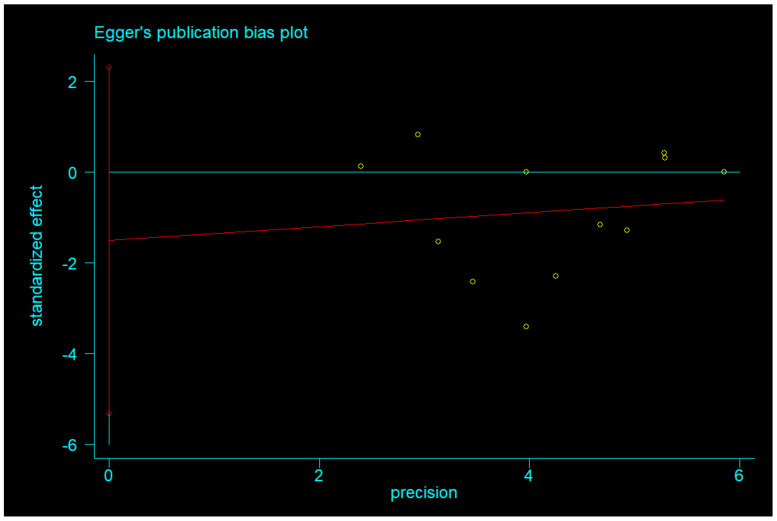
Egger’s publication bias plot of body weight.

**Figure 13 nutrients-14-01781-f013:**
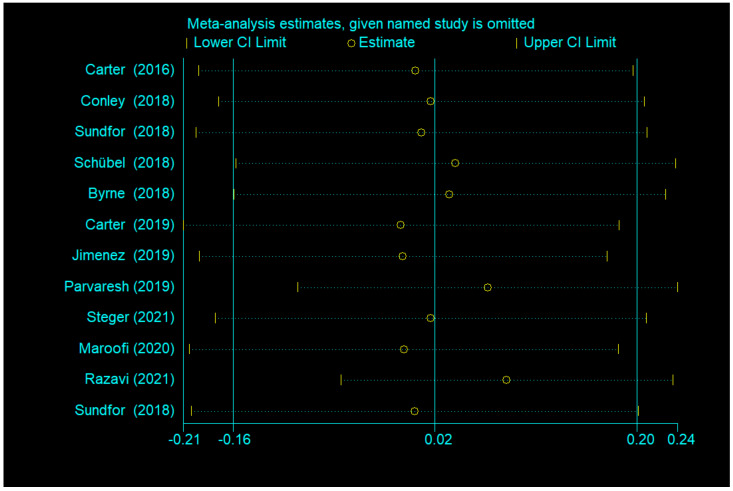
Impact analysis of a single study based on BMI.

**Figure 14 nutrients-14-01781-f014:**
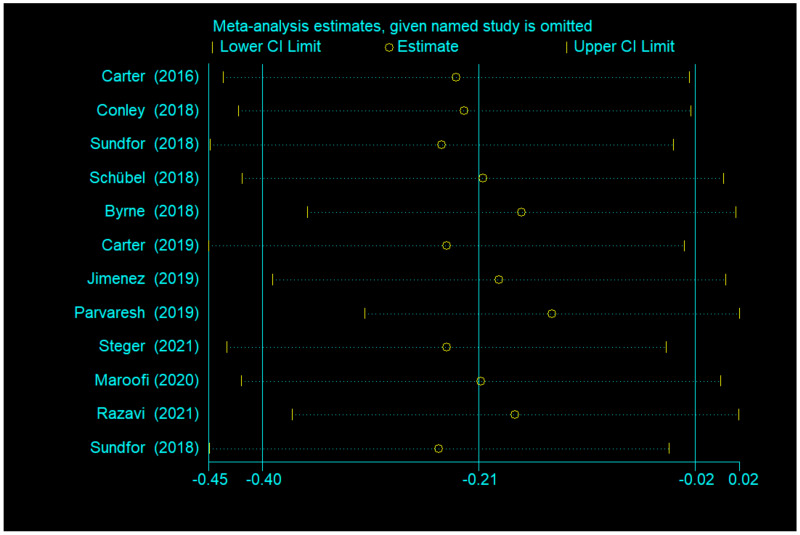
Impact analysis of a single study based on body weight.

**Table 1 nutrients-14-01781-t001:** Characteristics of included studies.

Author	Year	Country	N	Study Design	Participants	Study Duration	Interventions		Age, y	Body Weight, kg	BMI, kg/m^2^	Waist, cm
Carter et al.	2016	Australia	63	a pragmatic pilot trial	adults with overweight or obesity	12 weeks	IER vs. CER	IER	61 ± 7.5	99 ± 16	35 ± 4.8	-
								CER	62 ± 9.1	99 ± 15	36 ± 5.2	-
Conley et al.	2018	Australia	24	a randomised pilot stugy	obese war veterans	6 months	twice-weekly vs. CCR	twice-weekly	68 ± 2.7	99.1 ± 7.9	33.4 ± 1.8	114.2 ± 5.2
								CCR	67.1 ± 3.9	107.3 ± 17.1	36.2 ± 4.3	122.5 ± 10.4
Sundfor et al.	2018	Norway	112	RCT	adults with metabolic syndrome	6 months	IER vs. CER	IER	49.9 ± 10.1	108.6 ± 16.3	35.1 ± 3.9	116 ± 10
								CER	47.5 ± 11.6	107.5 ± 16.1	35.3 ± 3.5	116 ± 10
Schübel et al.	2018	Germany	150	RCT	overweight or obese nonsmokers	48 weeks	ICR vs. CCR	ICR	49.4 ± 9.0	96.4 ± 15.8	32.0 ± 3.8	-
								CCR	50.0 ± 8.0	92.5 ± 15.7	31.2 ± 4.0	-
Byrne et al.	2018	Australia	51	RCT	men with obesity	16 weeks	ICR vs. CCR	ICR	39.9 ± 9.2	109.8 ± 14.1	39.7 ± 6.8	-
								CCR	39.3 ± 6.6	111.6 ± 10.0	38.9 ± 5.2	-
Carter et al.	2019	Australia	137	RCT	adults with type 2 diabetes	12 months	IER vs. CER	IER	61 ± 9.0	100 ± 19	35 ± 5.8	-
								CER	61 ± 9.2	102 ± 17	37 ± 5.7	-
Jimenez et al.	2019	Spain	42	RCT	adults with overweight or obesity	6 weeks	ICR vs. CCR	ICR	46.32 ± 8.03	92.21 ± 13.82	32.83 ± 3.73	106.24 ± 11.89
								CCR	47.88 ± 7.67	97.99 ± 18.05	35.92 ± 5.32	110.49 ± 14.17
Parvaresh et al.	2019	Iran	70	RCT	adults with metabolic syndrome	8 weeks	ADF vs. CCR	ADF	44.6 ± 9.08	86.7 ± 10.65	31.1 ± 3.35	101 ± 9.41
								CCR	46.4 ± 7.94	84.2 ± 12.21	31.6 ± 3.82	103 ± 12.92
Steger et al.	2021	USA	35	RCT	adults with overweight or obesity	12 weeks	IER vs. CER	IER	43.4 ± 11	87.4 ± 11.5	31.1 ± 2.4	94.2 ± 8.8
								CER	48 ± 10	91.0 ± 9.7	31.4 ± 2.5	95.9 ± 9.3
Maroofi et al.	2020	Iran	88	RCT	subjects with overweight or obesity and mild-to-moderate HTG	8 weeks	ICR vs. CCR	ICR	44.0 ± 8.6	83.9 ± 13.7	31.6 ± 3.9	100.6 ± 9.8
								CCR	45.2 ± 11.7	90.1 ± 19.3	32.4 ± 4.6	104.7 ± 11.0
Razavi et al.	2021	Iran	80	RCT	adults with metabolic syndrome	4 months	ADF vs. CCR	ADF	41.3 ± 8.65	89.4 ± 7.72	31.3 ± 3.12	106 ± 9.71
								CCR	43.1 ± 9.26	87.1 ± 8.17	31.2 ± 3.95	104 ± 10.2

Data presented as means ± standard error of the mean. RCT, randomized clinical trials. HTG, hypertriglyceridemia. IER, intermittent energy restriction. CER, continuous energy restriction. ICR, intermittent calorie restriction. CCR, continuous calorie restriction. ADF, alternate-day fasting. BMI, body mass index.

**Table 2 nutrients-14-01781-t002:** Characteristics of included studies.

Author	Year	Country	Interventions		Fat Mass, kg	Percentage Fat Mass (%)	TC	TG	FBG	Insulin, IU/L	SBP, mmhg	DBP, mmhg	HOMA-IR
Carter et al.	2016	Australia	IER vs. CER	IER	38 ± 9.2	41 ± 7.9	-	-	-	-	134 ± 17	84 ± 10	-
				CER	40 ± 10.5	42 ± 7.7	-	-	-	-	138 ± 15	90 ± 11	-
Conley et al.	2018	Australia	twice-weekly vs. CCR	twice-weekly	-	-	3.9 ± 0.9 (mmol/L)	1.9 ± 0.6 (mmol/L)	-	-	141.5 ± 13.9	84.0 ± 9.5	-
				CCR	-	-	4.3 ± 1.0 (mmol/L)	2.4 ± 1.7 (mmol/L)	-	-	149.8 ± 18.3	88.1 ± 14.4	-
Sundfor et al.	2018	Norway	IER vs. CER	IER	-	-	4.87 ± 0.90 (mmol/L)	1.84 ± 0.83 (mmol/L)	5.8 ± 1.2 (mmol/L)	-	129 ± 13.4	88 ± 8.1	-
				CER	-	-	5.09 ± 0.87 (mmol/L)	1.55 ± 0.68 (mmol/L)	5.7 ± 0.7 (mmol/L)	-	128 ± 13.2	86 ± 8.7	-
Schübel et al.	2018	Germany	ICR vs. CCR	ICR	-	-	205.0 ± 30.8 (mg/dL)	130.0 ± 83.8 (mg/dL)	92.7 ± 7.5 (mg/dL)	11.6 ± 5.4	139.4 ± 18.7	136.0 ± 16.7	2.7 ± 1.3
				CCR	-	-	202.9 ± 39.3 (mg/dL)	121.2 ± 66.3 (mg/dL)	93.9 ± 7.5 (mg/dL)	12.6 ± 6.9	87.2 ± 9.9	87.3 ± 8.7	3.0 ± 1.7
Byrne et al.	2018	Australia	ICR vs. CCR	ICR	44.1 ± 11.5	39.7 ± 6.8	-	-	-	-	-	-	-
				CCR	43.6 ± 8.5	38.9 ± 5.2	-	-	-	-	-	-	-
Carter et al.	2019	Australia	IER vs. CER	IER	40 ± 9.4	42 ± 7.3	4.6 ± 1.3 (mmol/L)	1.5 ± 0.7 (mmol/L)	-	14 ± 20	-	-	-
				CER	42 ± 9.1	44 ± 6.6	5.0 ± 1.7 (mmol/L)	1.9 ± 1.4 (mmol/L)	-	14 ± 21	-	-	-
Jimenez et al.	2019	Spain	ICR vs. CCR	ICR	-	40.76 ± 6.61	-	-	-	-	-	-	-
				CCR	-	44.51 ± 6.40	-	-	-	-	-	-	-
Parvaresh et al.	2019	Iran	ADF vs. CCR	ADF	-	-	177 ± 36.52 (mg/dL)	199 ± 108.29 (mg/dL)	102 ± 9.17 (mg/dL)	13.07 ± 6.34	125 ± 9.78	84 ± 9.35	3.33 ± 1.69
				CCR	-	-	177 ± 37.17 (mg/dL)	218 ± 115.10 (mg/dL)	101 ± 7.58 (mg/dL)	14.28 ± 6.79	127 ± 14.03	83 ± 6.61	3.49 ± 1.86
Steger et al.	2021	USA	IER vs. CER	IER	37.9 ± 6.8	45.0 ± 4.7	-	-	-	-	119 ± 14	70 ± 11	-
				CER	41.5 ± 7.6	47.4 ± 6.3	-	-	-	-	123 ± 10	74 ± 10	-
Maroofi et al.	2020	Iran	ICR vs. CCR	ICR	-	37.5 ± 4.6	178.6 ± 30.3 (mg/dL)	180.5 ± 115 (mg/dL)	-	18.2 ± 8.1	-	-	3.5 ± 3
				CCR	-	35.9 ± 5.8	190.1 ± 38.1 (mg/dL)	165.0 ± 126 (mg/dL)	-	22.0 ± 9.7	-	-	3.7 ± 3
Razavi et al.	2021	Iran	ADF vs. CCR	ADF	37.1 ± 9.25	-	-	-	-	-	134 ± 9	86 ± 4	-
				CCR	34.2 ± 9.80	-	-	-	-	-	137 ± 10	85 ± 5	-

Data presented as means ± standard error of the mean. IER, intermittent energy restriction. CER, continuous energy restriction. ICR, intermittent calorie restriction. CCR, continuous calorie restriction. ADF, alternate-day fasting. TC, total cholesterol. TG, triacylglycerol. FBG, fasting blood glucose. SBP, systolic blood pressure. DBP, diastolic blood pressure.

**Table 3 nutrients-14-01781-t003:** Subgroups analyses of comparison of IF and CCR based on BMI changes.

	Groups	Participants	Random Effect SMD (95% CI)	I^2^ (%)	*p* for Heterogeneity
Over all	12	905	0.02 (−0.16, 0.20)	45	0.046
Subgroup analysis					
Age					
≥60 y	3	223	0.22 (−0.04, 0.49)	0	0.937
<60 y	9	682	−0.04 (−0.27, 0.18)	53.2	0.029
Area					
Oceania	4	274	0.15 (−0.09, 0.39)	0	0.651
Europe	4	364	0.10 (−0.11, 0.31)	5.5	0.366
Western Asia	3	232	−0.27 (−0.85, 0.31)	79.4	0.008
North America	1	35	0.08 (−0.59, 0.74)		
Physical condition					
Obesity or overweight with disease	6	593	0.05 (−0.18, 0.27)	0	0.568
Obesity or overweight	6	312	−0.03 (−0.32, 0.27)	68.9	0.007

**Table 4 nutrients-14-01781-t004:** Subgroups analyses of comparison of IF and CCR based on body weight changes.

	Groups	Participants	Random Effect SMD (95% CI)	I^2^ (%)	*p* for Heterogeneity
Over all	12	905	−0.21 (−0.40, −0.02)	48.1	0.031
Subgroup analysis					
Age					
≥60 y	3	223	0.01 (−0.26, 0.27)	0	0.993
<60 y	9	682	−0.28 (−0.52, −0.05)	56.6	0.018
Area					
Oceania	4	274	−0.15 (−0.48, 0.18)	38.9	0.178
Europe	4	364	−0.09 (−0.32, 0.14)	18.5	0.298
Western Asia	3	232	−0.53 (−0.87, −0.19)	41.1	0.183
North America	1	35	0.28 (−0.38, 0.95)		
Physical condition					
Obesity or overweight with disease	6	593	−0.21 (−0.48, 0.06)	27.3	0.23
Obesity or overweight	6	312	−0.22 (−0.50, 0.06)	64.8	0.014

## Data Availability

The datasets presented in this study can be found in online repositories. The names of the repository/repositories and accession number(s) can be found in the article.

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
