# Peer review of "Intermittent Fasting versus Continuous Calorie Restriction: Which Is Better for Weight Loss?"

_nutrients, 2022, doi:10.3390/nu14091781_

Round 1

Reviewer 1 Report

In the present review, Qing Zhang et al. investigated the effects of intermittent fasting versus continuous calorie restriction in the context of weight loss. The paper is quite interesting and original, although there is a lack of body composition analysis. Changes in body weight and BMI did not appear to be sufficient to compare the effects of CR and IF.

I suggest some changes below:

Line 55

There are comparisons between TRF and CR in the literature. Please correct the statement.

Line 56

On what basis do the authors question TRF as a form of IF? Please add references.

Line 65 and 73                                                                                                                                                 

Repeated sentences.

Line 80

The authors' research focused mainly on BMI, BW, WC, etc. I also propose to focus on BF, because weight changes after IF are rather small and may result from changes in body composition.

Discussion

I believe that when describing the specific action IF on the human body, authors should specify the form of IF. The different forms of IF show different effects, and describing all of them under the umbrella of IF seems to be confusing.

Reviewer 2 Report

The topic of this manuscript falls within the scope of Nutrients Journal. The topic of the manuscript is very interesting, relevant, and original.

It is a systematic review and meta-analysis of randomized clinical trials and pilot trial studies to compare the effectiveness of intermittent fasting (IF) and continuous calorie restriction (CCR) in overweight and obese people. The study aimed to compare IF and CCR regarding effectiveness for weight loss in people with obesity or metabolic syndrome.

The Authors have presented sufficient data. The appropriate tables and figures have been provided. The article is easy to read and logically structured.  The methods are adequately described. The Authors used appropriate statistic methods. The conclusions are consistent with presented evidence and arguments.

the strength of this paper: very interesting topic; material and methods-the right choice of methodology methods, which was presented in comprehensible way; the obtained results are presented in the form of figures and tables, which are clear and easy to understand; the discussion- supports the results properly and refers to the current literature in appropriate manner; the conclusions- based on the obtained results, they are consistent with evidence and arguments. They address the main question posed.  The Authors used appropriate references.

There are some comments in the reviewer opinion which should be taken under consideration by the Authors:

1.In the introduction please include that obesity also has an impact on perivascular adipose tissue (PVAT) (please cite PMID: 34836100, PMID: 27761903)

2.Please include in introduction that “Obesity triggers alterations in the quantity and quality of various types of cells that reside in adipose tissue, including adipose stem cells (ASCs). These alterations in the functionalities and properties of ASCs impair adipose tissue remodeling and adipose tissue function, which induces low-grade systemic inflammation, progressive insulin resistance, and other metabolic disorder” ( please cite https://doi.org/10.3390/nu14071509; DOI: 10.1042/BSR20194076)

3.Please add side effects of these diet interventions

4.Please add a section on future directions
